# Quiescence unveils a novel mutational force in fission yeast

**Serge Gangloff[1,2†], Guillaume Achaz[3†], Stefania Francesconi[1,2], Adrien Villain[1‡], Samia Miled[1,2], Claire Denis[1,2], Benoit Arcangioli[1,2]\***

[1]Genomes and Genetics, Institut Pasteur, Paris, France; [2]UMR 3525, CNRS-Institut Pasteur, Paris, France; [3]ISYEB UMR7505 CNRS MNHN UPMC EPHE CIRB UMR 7241 CNRS Collège de France INSERM, UPMC, Paris, France

**Abstract** To maintain life across a fluctuating environment, cells alternate between phases of cell division and quiescence. During cell division, the spontaneous mutation rate is expressed as the probability of mutations *per* generation (Luria and Delbrück, 1943; Lea and Coulson, 1949), whereas during quiescence it will be expressed *per* unit of time. In this study, we report that during quiescence, the unicellular haploid fission yeast accumulates mutations as a linear function of time. The novel mutational landscape of quiescence is characterized by insertion/deletion (indels) accumulating as fast as single nucleotide variants (SNVs), and elevated amounts of deletions. When we extended the study to 3 months of quiescence, we confirmed the replication-independent mutational spectrum at the whole-genome level of a clonally aged population and uncovered phenotypic variations that subject the cells to natural selection. Thus, our results support the idea that genomes continuously evolve under two alternating phases that will impact on their size and composition.

DOI: https://doi.org/10.7554/eLife.27469.001

**\*For correspondence:** barcan@pasteur.fr

[†]These authors contributed equally to this work

**Present address:** [‡]Laboratoire Information Génomique & Structurale, CNRS, Marseille, France

**Competing interests:** The authors declare that no competing interests exist.

## Introduction

The causes and consequences of spontaneous mutations have been extensively explored. The major sources include errors of DNA replication and DNA repair and the foremost consequences are genetic variations within a cell population that can lead to heritable diseases and drive evolution. The knowledge of the rate and spectrum of spontaneous mutations is also very informative and of fundamental importance to understand their origin. During cell division, fluctuation assay (*Luria and Delbrück, 1943*; *Lea and Coulson, 1949*) and more direct measurements using next generation sequencing, including mutation accumulation (MA) (*Halligan and Keightley, 2009*) and de novo mutations (*Conrad et al., 2011*) improved the mutation rate estimations expressed as the number of spontaneous mutations *per* cell division and led to the definition of their spectrum in many species (*Behringer and Hall, 2015*; *Farlow et al., 2015*; *Zhu et al., 2014*). During growth, most of the mutations are due to DNA replication errors. When the mutation is neutral or beneficial, it can be fixed in the population during subsequent generations. Conversely, when the mutation is disadvantageous it most often rapidly counter selected. The replication-dependent mutations define a spectrum quite similar among species, with the domination of Single Nucleotide Variants (SNVs) over Insertions/ Deletions (Indels) and Structural Variants (SVs). For instance, preference for insertions at the expense of deletions along with a universal substitution bias toward AT has been frequently reported during cell division (*Hershberg and Petrov, 2010*; *Lynch, 2010*). At the evolutionary timescale, the mutations accumulate as a linear function of time (*Zuckerkandl and Pauling, 1962*). However, the mutation rate is not constant and depends on the generation time, the efficiency of the DNA damage protection, the accuracy of DNA repair and the selective environment. In the past decades, it has become increasingly evident that mutations also arise during cell cycle arrest, slow growth or under

stress. Many genetic studies on *E. coli* and budding yeast used the term 'adaptive mutation' (*Drake, 1991*; *Foster, 1999*) as they used non-lethal selective conditions for an essential amino acid, nucleotide or antibiotic. An important notion related to adaptive mutation is that stress conditions may increase mutations and trigger accelerated evolution (*Yaakov et al., 2017*; *Long et al., 2016*; *Rosenberg, 1997*; *Hicks et al., 2010*; *Holbeck and Strathern, 1997*). A more recent notion is that a bacterial subpopulation of phenotypic variants called 'persisters' are more resistant to stress conditions suggesting that they precede adaptive mutations (for review (*Harms et al., 2016*)), a notion that was also found in budding yeast (*Yaakov et al., 2017*). The survival of 'persisters' to a large range of stress conditions can be explained by a reduced growth rate and metabolism. In addition, life alternates between periods of cell division and quiescence. During quiescence, the main replication-dependent source of mutations is not applicable but others remain, such as DNA repair errors, that may potentially improve the chance of survival. In other words, the respective fitness of cell division and quiescence might alternatively subject organisms to natural selection. In this regard, the extent of the impact of replication-independent mutations on the overall mutation load and evolution is mostly unknown (*Gao et al., 2016*). Hence, because of its mechanistic difference, a replication-independent mutational spectrum is expected to exhibit a different signature.

Quiescence is a common cell state on earth (*Lewis and Gattie, 1991*). In metazoan, stem cells alternate between variable periods of growth and quiescence depending on the period of development and the type of tissues. For the unicellular fission yeast, *Schizosaccharomyces pombe*, removing nitrogen triggers mating of opposite mating-types followed by meiosis. However, when the population is composed of only one mating-type they arrest in the G1-phase and rapidly enter into quiescence with a 1C content (*Nurse and Bissett, 1981*). In these conditions, the cells remain viable for months given the medium is refreshed every other week. They are metabolically active, exhibit stress-responsive signaling and are highly efficient in DNA damage repair (*Mochida and Yanagida, 2006*; *Yanagida, 2009*; *Ben Hassine and Arcangioli, 2009*; *Marguerat et al., 2012*; *Gangloff and Arcangioli, 2017*). Thus, quiescence in fission yeast is defined under a simple nutritional change so that studies can be reproduced and interpreted rigorously.

Here, we report the accumulation and spectrum of spontaneous mutations that arise in the quiescent phase of fission yeast. The growth and quiescence mutational spectra exhibit quantitative and qualitative differences that further explore the genetic potential of the genome. We named the new quiescence mutational spectrum 'Chronos' the personification of time in Greek mythology.

## Results

In all our experiments, a prototrophic progenitor fission yeast strain is grown in minimal medium (MM) (*Mitchison, 1970*) prior to transfer into MM lacking nitrogen at a cell density of $10^6$ cells per milliliter. After two divisions, a majority of the cells arrests in the G1 phase and enters into the G0 quiescent state. To determine the homogeneity of the cell population during quiescence, we analyzed the size and the proportion of cells exhibiting a septum during 15 days (*Table 1*). The cell-size (7–14 μm) and proportion of cells with a septum (~10%) observed during vegetative growth abruptly decrease after 1 day of quiescence, with more than 99,9% of septum free cells measuring 4 μm. During the following days of quiescence, the proportion of cells containing a septum dropped below 1

**Table 1.** Proportion of septa observed among cycling and quiescent cells.

| Days | Number of cells | Number of Septa | Percentage |
|------|-----------------|-----------------|------------|
| 0 | 1 400 | 136 | 9.71% |
| 1 | 3 500 | 5 | 0.14% |
| 4 | 10 500 | 0 | 0.00% |
| 6 | 16 000 | 2 | 0.01% |
| 8 | 14 000 | 0 | 0.00% |
| 12 | 26 000 | 0 | 0.00% |
| 15 | 26 250 | 0 | 0.00% |

DOI: https://doi.org/10.7554/eLife.27469.006

in 20,000 cells. This result indicates that the proportion of cells dividing or replicating during quiescence is very low. The efficiency and accuracy of the repair of DNA lesions in quiescence remain unknown, and lesions can be converted into mutations either during quiescence or when cells re-enter the vegetative cycle. We determined the mutation frequencies by plating samples of quiescent cultures after 1, 4, 8, 11 or 15 days in MM lacking nitrogen onto rich medium containing 5-fluoroorotic acid (5-FOA) that allows the recovery of ura4- and ura5- loss-of-function mutants (*Grimm et al., 1988*). We ascertained that the *ura4Δ* mutants remain viable for two weeks of quiescence, indicating that colonies resistant to 5-FOA (FOA[R]) emerging early are not biased by selection (*Figure 1*) and that our phenotypic accumulation assay is unbiased during the course of the experiment. FOA[R] colonies were scored and their DNA isolated for mutational spectrum analysis by Sanger sequencing. At day 1, a large fraction of FOA[R] colonies derive from replicative mutations that have appeared during the last rounds of DNA replication prior or during entry into quiescence and are capable of surviving two to three generations on media lacking uracil. Because of the fluctuation caused by the timing of mutation appearance in lineages (*Luria and Delbrück, 1943*), a mutation can be found multiple times in a population. To assess the mutational spectra in growth and quiescence, mutations found

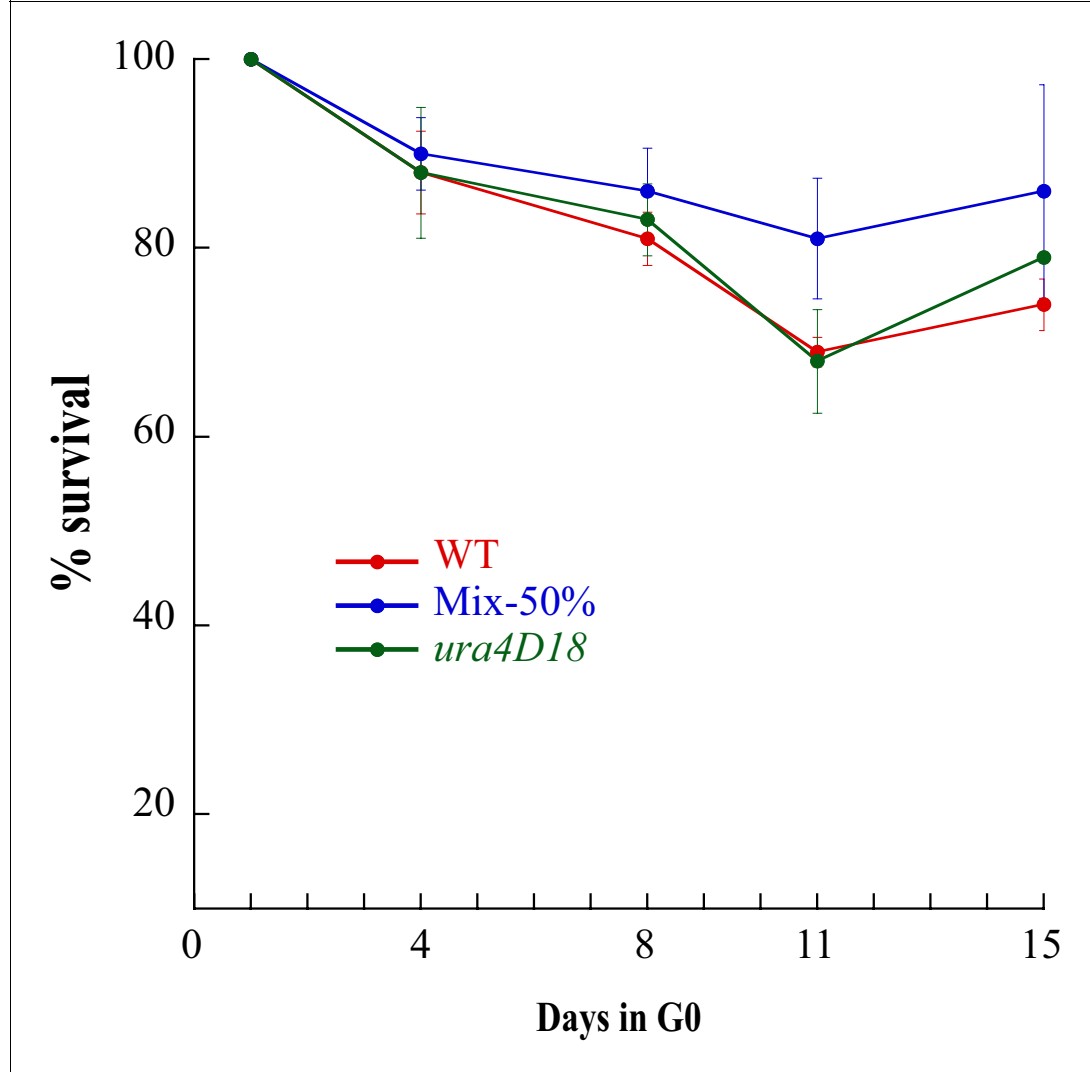

**Figure 1.** Survival of ura- cells during two weeks of quiescence. Wild-Type, *ura4D18* cells and a 50% mixture of both cultures were put into quiescence and their survival was followed for two weeks. *ura4D18* has no advantage compared to wild-type strain or mixed populations and the proportion of uracil auxotrophs is maintained constant for the two weeks.
DOI: https://doi.org/10.7554/eLife.27469.002

more than once in a clonal population were discarded (*Supplementary files 1*). The frequency of non-redundant FOA$^R$ mutations at day 1 ranges from 1 to $3 \times 10^{-7}$ across the various independent experiments (*Figure 2A*). We next analyzed their spectrum at day one by Sanger sequencing the respective *ura4*$^-$ or *ura5*$^-$ mutated gene. If every substitution occurs with an equal probability, we should observe one transition per two transversions. We found a 1:2.18 ratio, with a mutational bias towards the enrichment of A/T (3.71, *Table 2*) due to the high frequency of C:G to T:A transitions and G:C to T:A transversions (*Figure 2B* and *Figure 2—figure supplement 1*). Interestingly, as previously observed (*Behringer and Hall, 2015*; *Farlow et al., 2015*; *Zhu et al., 2014*) the CpG dinucleotides are found among the top mutated dinucleotides in *ura4* and *ura5* (*Figure 2—figure supplement 2*) a feature difficult to understand in the absence of cytosine methyl transferase in fission yeast. Among the *ura4*$^-$ and *ura5*$^-$ FOA$^R$ mutations, 72% are caused by Single Nucleotide Variants (SNVs) and 28% by insertions/deletions (indels) (*Figure 2C*). A slight bias for insertions is

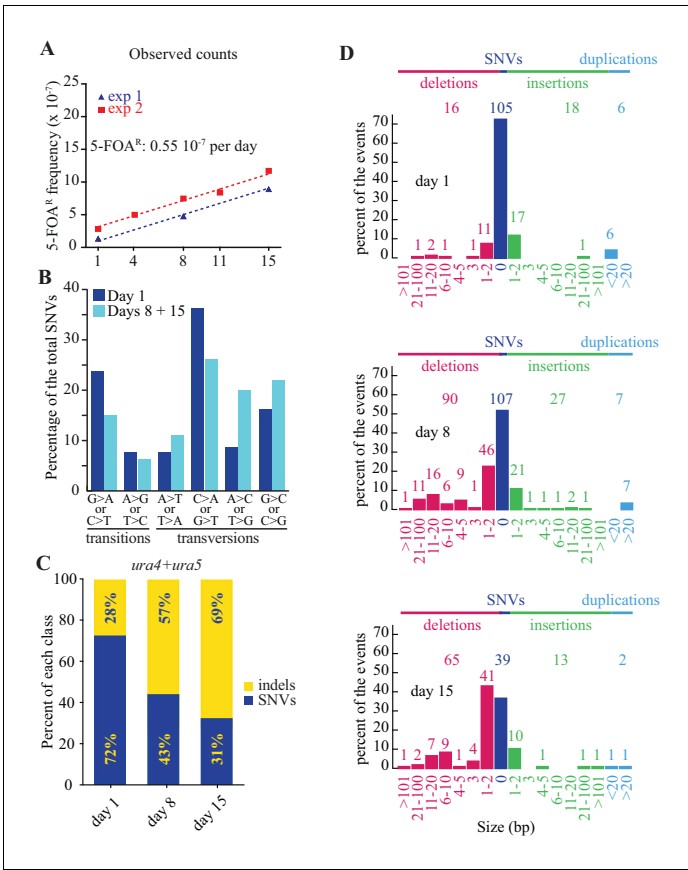

**Figure 2.** Mutations accumulate as a function of time in quiescence. (**A**) slopes of FOA$^R$ accumulation as a function of time in quiescence from two independent experiments, determined by the least squares regression ($R^2 = 0.99$ for all mutations, $R^2 = 0.96$ for SNVs and $R^2 = 0.80$ for indels). (**B**) mutation spectrum established on 105 SNVs found in the *ura4*$^+$ and *ura5*$^+$ genes at day 1 and 146 at days 8 + 15. (**C**) distribution among indels and SNVs of the mutations that result in FOA$^R$ for *ura4*$^+$ and *ura5*$^+$ genes over time. (**D**) distribution of the various sizes of indels over time in quiescence. The numbers directly above the histograms indicate the number of events, while those on the top of the figure are the sum of all the events for each class. The numbers at the bottom indicate the size range of the indels.

DOI: https://doi.org/10.7554/eLife.27469.003

The following figure supplements are available for figure 2:

**Figure supplement 1.** SNVs distribution.
DOI: https://doi.org/10.7554/eLife.27469.004

**Figure supplement 2.** Normalized percentage of each dinucleotide in the open reading frames of *ura4*$^+$ and *ura5*$^+$ after one day in quiescence.
DOI: https://doi.org/10.7554/eLife.27469.005

**Table 2.** Type of mutation and AT bias found in *ura4* and *ura5* mutants at various time points of quiescence.

|  | day 1 | day 8 | day 15 | day 8+15 |
|---|---|---|---|---|
| # of SNVs | 105 | 107 | 37 | 144 |
| Ts | 33 | 26 | 5 | 30 |
| Tv | 72 | 81 | 34 | 116 |
| Ts:Tv | 1:2.18 | 1:3.12 | 1:6.67 | 1:3.70 |
| To A or to T | 63 | 45 | 15 | 60 |
| To C or to G | 17 | 30 | 8 | 38 |
| AT bias | 3.71 | 1.50 | 1.88 | 1.58 |
| # of indels | 40 | 139 | 86 | 220 |
| # of deletions | 16 | 99 | 71 | 170 |
| lost bases | 334 | 864 | 907 | 1771 |
| # of insertions | 24 | 35 | 15 | 50 |
| gained bases | 269 | 401 | 99 | 500 |
| net loss | 65 | 463 | 808 | 1271 |
| # of deletions / # of insertions | 0.7 | 2.8 | 4.7 | 3.4 |
| average loss per event | 21 | 9 | 13 | 10 |
| average gain per event | 11 | 11 | 7 | 10 |

Ts: transitions; Tv: transversions

AT bias: (GC→AT + GC→TA) to (AT→GC + AT→CG) mutations

DOI: https://doi.org/10.7554/eLife.27469.007

observed with a net gain of 269 bp for 24 events and loss of 334 bp for 16 events (*Table 2*), including two deletions of 165 and 95 base-pairs (*Supplementary files 1*). Overall, the mutation profile at day one is similar to published results in cycling cells for *URA3* in budding yeast (*Lang and Murray, 2008*) and for *ura4*[+] and *ura5*[+] in fission yeast (*Fraser et al., 2003*).

During quiescence, the total number of mutations (including redundant ones) resulting in FOA[R] colonies increases linearly as a function of time. From multiple experiments, we used least squares regression to determine that the slope is $0.55 \times 10^{-7}$ FOA[R] mutants per day spent in quiescence (*Figure 2A*). Importantly, we observed that the number of redundant mutations dramatically fades over time, indicating that novel mutations arise in quiescence (*Supplementary files 1*). Days 8 and 15 are not statistically different for the Ts:Tv and AT bias (chi2 test) and were therefore combined. Day 1 is statistically different from days 8 + 15 (Ts/Tv: p<0.05; AT bias: p<0.05 by chi2 tests). The ratio of transition-to-transversion observed at days 8 + 15 of quiescence was reduced from 1:2.18 to 1:3.7 and the mutational bias toward A/T decreased (3.71 vs 1.58, *Table 1*). This is mainly due to a relative decrease of the G:C to A:T transitions that is balanced by an increase of the A:T to C:G and G:C to C:G transversions (*Figure 2B* and *Figure 2—figure supplement 1*). During two weeks of quiescence, the proportion of de novo indels increases from 28% at day 1 to 57% and 69% after 8 and 15 days, respectively (*Figure 2C*) to outnumber the SNVs (*Figure 2C and D*) (day 1 *vs* days 8 + 15: p<10$^{-10}$, chi2 test). Altogether, we found more deletions than insertions (in days 8 + 15: p<10$^{-5}$, chi2 test) with a net loss of 1771 bp in 170 mutants and a gain of 500 bp in 50 mutants (*Figure 2D*, *Table 2*). The main class of indels is ±1 bp and accounts for roughly half the events (*Figure 2D*). More than one half of the indels occur within homonucleotide runs and several mutations are complex (*Supplementary files 1*). Collectively, we found a phenotypic mutation rate of $0.55 \times 10^{-7}$ FOA[R] colonies per day of quiescence with $0.14 \times 10^{-7}$ FOA[R] colonies per day due to SNVs and $0.41 \times 10^{-7}$ FOA[R] colonies per day due to indels (*Figure 3A,B*). Thus, the mutational spectra of growth and quiescence exhibit striking quantitative and qualitative differences. First, elevated levels of indels are generated. Second, two recent MA studies (*Behringer and Hall, 2015*; *Farlow et al., 2015*) with fission yeast have shown that in cycling cells insertions outnumber deletions, whereas we observe the reverse in quiescence (*Figure 2D*). Thus, growth and quiescence apply opposite

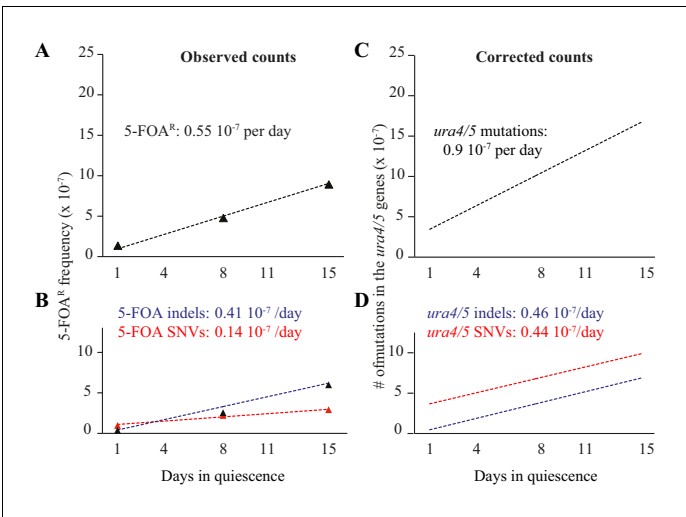

**Figure 3.** FOA[R] accumulation and corrected counts. (**A**) slope of FOA[R] (as determined in *Figure 2A*). (**B**) slopes of FOA[R] SNVs and indels accumulation. (**C**) corrected counts determined on all experimental points using least squares regression (**D**) corrected slopes for SNVs and indels formation in the *ura4*[+] and *ura5*[+] genes.
DOI: https://doi.org/10.7554/eLife.27469.008

pressures on the *S. pombe* genome size. Third, during cell divisions, SNVs elevate the genomic A/T composition (*Hershberg and Petrov, 2010*; *Lynch, 2010*). This bias is reduced during quiescence and counteracts the universal A/T enrichment observed during cell division. This suggests that, in addition to the previously recognized meiotic recombination, gene conversion, nucleotide modifications or transposition, the equilibrium of size and composition of the *S. pombe* genome also depends on the relative strength of the opposing forces applied during growth and quiescence.

A possible caveat of our study is that FOA is known to perturb the metabolism of uracil and thymidine and might thus impact differently the mutational spectra of growing and quiescent cells. To evaluate this issue, a yeast cell population was maintained during 6 days in quiescence, and one half of the culture (1.2 10[8] cells) was directly plated onto FOA plates while the other half was allowed to divide once in rich medium prior to plating. Note that we plated and sequenced all the FOA[R] mutants present in the culture, allowing to identify all the mutations. The experiment was performed after six days of quiescence when the proportion of quiescent mutations is elevated and the exit of quiescence is still synchronous. We obtained (i) 27 FOA[R], including 8 SNVs and 19 Indels when quiescent cells are directly plated on FOA (ii) 56 FOA[R] when cells are allowed to divide once (iii) 40 mutations appeared as pairs of mutations, including 2 × 9 SNVs and 2 × 11 Indels and 16 unique mutations, including 7 SNVs and 9 Indels (iv) All mutations exhibit the quiescence mutational profile, indicating that the FOA drug does not significantly impact on the mutation spectrum of quiescent or dividing cells (*Table 3*). The 16 unique mutations suggest a frequent heterozygosity of the mutations that can be revealed only after cell division.

**Table 3.** Sequencing analysis of all the FOAR colonies recovered after 6 days of quiescence.

|  |  | SNVs | Indels | Insertions | Deletions |
|---|---|---|---|---|---|
| No division | Singles | 8 | 19 | 2 | 17 |
|  | Pairs | NA | NA | NA | NA |
| One division | Singles | 7 | 9 | 3 | 6 |
|  | Pairs | 9 | 11 | 2 | 9 |

NA: Not Applicable. All the FOA[R] isolated from the culture have been sequenced.

For the first half of the cultures, the mutations found more than once in the clonal population were discarded.

Similarly, identical mutations found in 'no division' and in 'with one division' were discarded.

DOI: https://doi.org/10.7554/eLife.27469.009

We have developed two new methods based on a likelihood framework to infer the total mutation count from the measured 5-FOA$^R$ phenotypic mutation rate (methods 1 and 2). We have also computed the total mutation count from a previously proposed method (here method 3) (*Drake, 1991*; *Lang and Murray, 2008*), (Materials and methods). For the sake of clarity, we report the corrections obtained using method 1, but provide in parenthesis inferences from the two other methods. When applied to the *ura4$^+$/5$^+$* genes together, we found that the corrected number of mutations per day is $0.93 \times 10^{-7}$ taking into account the mutations that do not result in a 5-FOA$^R$ phenotype ($1.86$ or $1.24 \times 10^{-7}$ from methods 2 or 3). When applied to SNVs and indels, we found very similar slopes of $0.46 \times 10^{-7}$ and $0.47 \times 10^{-7}$, respectively (*Figure 3D*, *Table 4*) ($1.27{:}0.59$ and $0.50{:}0.74 \times 10^{-7}$ using methods 2 and 3). Next, we used this method to extrapolate the number of mutations in the quiescent genome. We expected to find $0.93 \times 10^{-7}/1400$ (nucleotides of *ura4 +- ura5*) x $14 \times 10^6$ (nucleotides in the *S. pombe* genome)$=0.93 \times 10^{-3}$ mutations per genome per day ($1.86$ and $1.24 \times 10^{-3}$ using methods 2 and 3).

To extend our observations made on FOA$^R$ mutations to the whole genome, we analyzed the mutation spectrum in cells that survived for 3 months of quiescence. For long-term experiments, we changed the medium every other week to maintain oligo elements and glucose as well as to prevent the survivors to feed on the nitrogen released by dead cells. In these conditions, the viability at 3 months is about 0.05% (*Figure 4*). From several experiments, we observed a biphasic viability curve, with a cell death acceleration after three weeks of quiescence (*Figure 4*). DNA from 243 colonies was purified. Illumina libraries were constructed and paired-end sequenced with an average

**Table 4.** Estimation of the fraction of non-synonymous SNVs that lead to a phenotype.

| | | *ura4* | *ura5* | Total |
|---|---|---|---|---|
| | | (265 Aa) | (216 Aa) | (481 Aa) |
| Potential SNVs | Synonymous | 530 (0.22) | 443 (0.23) | 973 (0.22) |
| | Non-synonymous | 1739 (0.73) | 1405 (0.72) | 3144 (0.73) |
| | STOPs | 116 (0.05) | 96 (0.05) | 212 (0.05) |
| Observed SNVs | Synonymous | 3 (0.03) | 2 (0.03) | 5 (0.03) |
| | Non-synonymous | 74 (0.69) | 58 (0.73) | 132 (0.71) |
| | STOPs | 29 (0.27) | 19 (0.24) | 48 (0.26) |
| | Non-coding | 1 (0.01) | 0 (0.00) | 1 (0.00) |
| | Different mutations per Aa | (41,13,3) | (23,7,5,0,0,1) | (64,20,8,0,0,1) |
| | Independent identical mutations per non-syn SNV | (64,9,1) | (41,11,2,1,1,0,1, 0,1) | (105,20,3,1,1,0,1, 0,1) |
| Method 1 | Essential Aa (ML) | 124 | 54 | 170 |
| (Essential Aa) | [CI 95%] | [93,202] | [43,81] | [140,228] |
| | *f* | 0.47 | 0.25 | 0.35 |
| | [CI 95%] | [0.35,0.76] | [0.20,0.38] | [0.29,0.47] |
| Method 2 | Essential NS (ML) | 296 | 88 | 276 |
| (Non-Syn) | [CI 95%] | [195,394] | [73,118] | [228,362] |
| | *f* | 0.17 | 0.06 | 0.09 |
| | [CI 95%] | [0.11,0.23] | [0.05,0.08] | [0.07,0.12] |
| Method 3 | # STOPs among all potentials STOPs | 0.25 | 0.20 | 0.23 |
| (LM08) | *f* | 0.17 | 0.21 | 0.19 |

DOI: https://doi.org/10.7554/eLife.27469.010

The following source data available for Table 4:

Source data 1. Estimation of the fraction of SNVs and Indels that leads to a phenotype.
DOI: https://doi.org/10.7554/eLife.27469.011

Source data 2. Estimated mutation rates.
DOI: https://doi.org/10.7554/eLife.27469.012

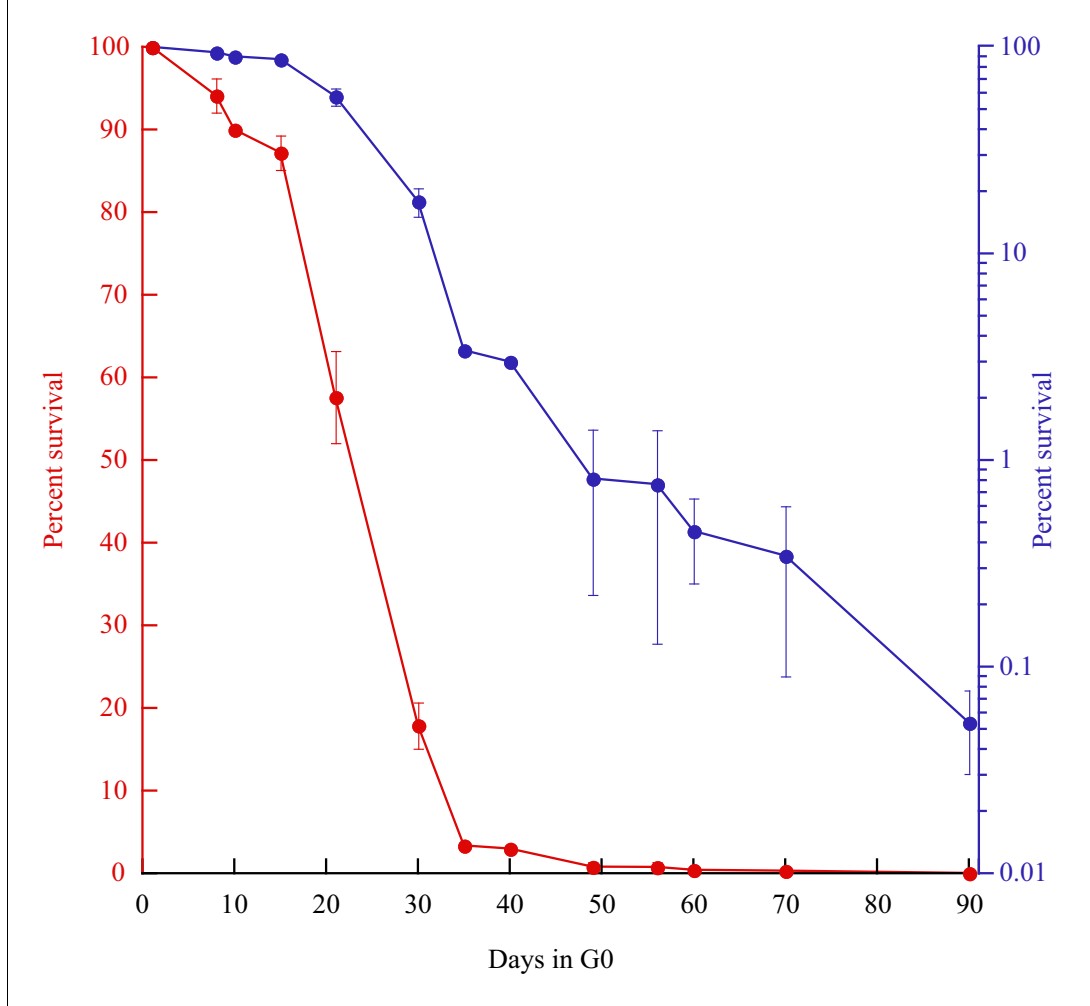

**Figure 4.** Survival curves (red: linear; blue: log) of prototrophic cells in G0 as a function of time. The medium is being replaced every other week starting at day 15, while an aliquot is plated out to monitor the viability. The standard error of the experiments is indicated.
DOI: https://doi.org/10.7554/eLife.27469.013

coverage above 50x to maintain high quality sequences and a low false discovery rate (FDR) that was experimentally validated (see Material and methods). SNVs and short indels were determined using Genome Analysis Toolkit (GATK) (*McKenna et al., 2010*; *DePristo et al., 2011*; *Van der Auwera et al., 2013*), and we combined the output of several tools including SOAPindel, Prism and Pindel (*Li et al., 2013*; *Jiang et al., 2012*; *Ye et al., 2009*) to increase the sensitivity of indels detection. We performed a stringent calling procedure for both SNVs and indels and we only considered variants that are present in at least 40% of the reads with a local coverage above 10x. Sanger sequencing was used to validate the de novo variants and to estimate the FDR.

We report 149 unique mutations, including 72 SNVs and 77 indels from the 243 sequenced genomes (*Table 5* and *Supplementary files 2*). Although low, 0.61 mutations per genome after three months, is 3.5–7 times higher than anticipated by our projection (0.084 to 0.17 mutations/genome/three months, depending on the correction method), indicating that this process may not be linear for extended periods of time, as suggested by the viability curve (*Figure 4*). Among the 59 SNVs involved in the AT bias, we found an AT/GC ratio bias of 1.27, a value intermediates between 1.84 at day 1 or 1.1 at days 8 + 15. This value is higher than those observed in our targeted experiments, but lower than in cycling cells. Among the 77 indels, all the events detected with GATK were found as well with either Pindel, SOAPindel or Prism (*Li et al., 2013*; *Jiang et al., 2012*; *Ye et al., 2009*). Among these indels, 26 are deletions (25/77–35%), a value two times greater than reported

**Table 5.** Type of mutation in whole genomes after 3 months in quiescence.

| Variants | # |
|---|---|
| # of SNVs | 72 |
| Ts | 26 |
| Tv | 46 |
| Ts:Tv | 1:21.77 |
| to A or T | 33 |
| to C or G | 26 |
| AT bias | 1.27 |
| # of indels | 77 |
| # of deletions | 27 |
| lost bases | 511 |
| # of insertions | 50 |
| gained of bases | 296 |
| net loss | 215 |
| # of deletions / # of insertions | 0.54 |
| average loss per event | 19.6 |
| average gain per event | 5.92 |

DOI: https://doi.org/10.7554/eLife.27469.016

in MA lines of cycling cells (14%–17%) (**Behringer and Hall, 2015**; **Farlow et al., 2015**). Consistent with what was observed in the FOA[R] study, over 62% (48/77) of the insertions and deletions are ±1 nucleotide. With respect to genome size, the 77 indels led to the net loss of 215 (511-296) nucleotides. We also analyzed the distribution of the number of mutations (143) per genome (237) that fits perfectly a Poisson distribution. We removed from the calculation the six mutated strains that were screened for phenotypic analysis, see below (**Figure 5**). This result shows that a unique mutational force, equally affecting all genomes, is at work during quiescence. The long-term quiescence experiment results support (i) the comparable amounts of indels and SNVs predicted in our estimation (**Figure 6**) (ii) that the proportion of deletions among indels is higher than in MA lines of cycling cells, with more nucleotides lost than gained (iii) over 62% of the deletions/insertions are ±1 events. Altogether, quiescence in fission yeast reveals Chronos as a new genetic force generating similar proportion of SNV and Indels, together with an enrichment of deletions.

The level of mutations that we have observed after three months in quiescence is likely to cause some heritable phenotypic diversity. Therefore, we conducted a phenotypic survey in conditions that affect a broad range of cellular functions. We did not observe any phenotype upon examination of 384 colonies after 1 day or 1 month of quiescence. However, after 2 and 3 months, we observed 4/376 (1.1%) and 6/334 (1.8%) colonies displaying phenotypes (**Table 6**), respectively. Genetic crosses confirmed that the phenotype observed in the 10 colonies derives from a single mutated locus. We sequenced the genomes of the six strains isolated after three months and found mutations mapping to *alg5*, *alg8*, *aur1*, *ccc2*, *gdi1* and *oca8* genes. These genes encode for cell wall and vacuole/membrane trafficking functions that might improve viability during quiescence. Taken together, we conclude that cellular quiescence allows for genetic variation.

## Discussion

Extensive work has shown that quiescence is a well-controlled and conserved process (**Yanagida, 2009**; **Lee et al., 1988**; **Petersen and Hagan, 2005**). Two recent studies on RNA interference and telomere stability have highlighted the importance of quiescence for chromosome biology (**Maestroni et al., 2017**; **Roche et al., 2016**). Our experimental conditions generate a homogenous quiescent cell population that remains stable during the mutation accumulation experiments. However, we cannot exclude that a very low proportion (<10$^{-4}$) of cells are able to divide during

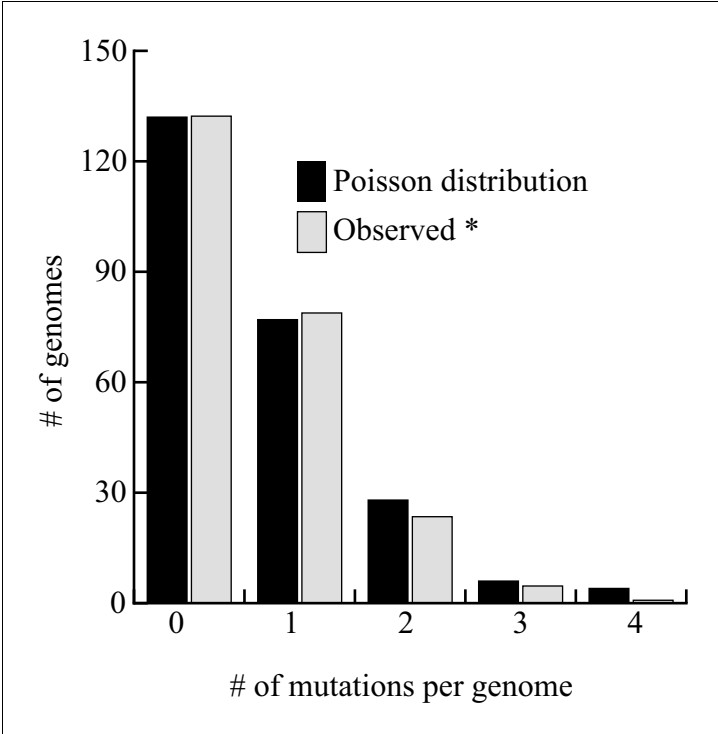

**Figure 5.** Distribution of mutations among the 237 genomes sequenced. The six strains exhibiting a phenotype were excluded from the analysis. The fit between the observed and theoretical Poisson distribution indicates that Chronos acts homogeneously on genomes over time. The goodness of fit is excellent (Pchi2 = 0.56).
DOI: https://doi.org/10.7554/eLife.27469.014

quiescence. For instance, it is possible that persister cells, as observed during vegetative growth in budding yeast (*Yaakov et al., 2017*), will not arrest in nitrogen starvation conditions and attempt to replicate and divide later on during quiescence. If these replicating cells were generating the FOA[R] mutants observed during quiescence, we will have to conclude that they generate 1000 times more mutations per replication. Double-strand break (DSB) repair is a process that could generate this level of mutations (*Hicks et al., 2010*; *Holbeck and Strathern, 1997*). Nevertheless, these rare replicating cells will have to divide at a constant rate during time in quiescence. Additionally, the number and the distribution of mutations per genome observed in the three months aged cells is hardly compatible with this hypothesis. Therefore, we favor the Chronos model where DNA lesions (not excluding DSBs) that are arising during quiescence (*Mochida and Yanagida, 2006*; *Ben Hassine and Arcangioli, 2009*) are repaired with errors that generate mutations with time.

If the mutations are not generated by errors during DNA replication, what are the mechanism(s) underlying Chronos? As a first attempt, we investigated the involvement of the translesion DNA synthesis (TLS) pathway. We found that the *Pcn1-K164R* mutant (*Coulon et al., 2010*) that cannot switch from replicative to TLS mode accumulates only slightly more of FOA[R] mutants than *wt* and displays the Chronos profile (*Figure 7*). Thus, the participation of the TLS pathway in the mutational signature of quiescence is modest. Further investigations of the DNA repair pathways at work during quiescence are needed not only to provide information on the importance of each DNA repair pathway but also valuable insights into physiological source of the spontaneous DNA lesions in quiescence.

Initially, Luria and Delbruck (*Luria and Delbrück, 1943*) expressed the mutation rate *per unit time*, that was transformed to *per generation* later by Lea and Coulson (*Lea and Coulson, 1949*). Now, the common assumption is that the overall mutation rate and spectrum accompanying cell division is combining errors occurring during DNA replicating and/or repair. It was recently proposed that inefficiently repaired lesions increase according to absolute time (*Gao et al., 2016*). Our work proposes that quiescence is a genetic active state that participates to the mutation fraction that is expressed in unit of time. Thus, *S. pombe* genome fluctuates between two mutational spectra, which

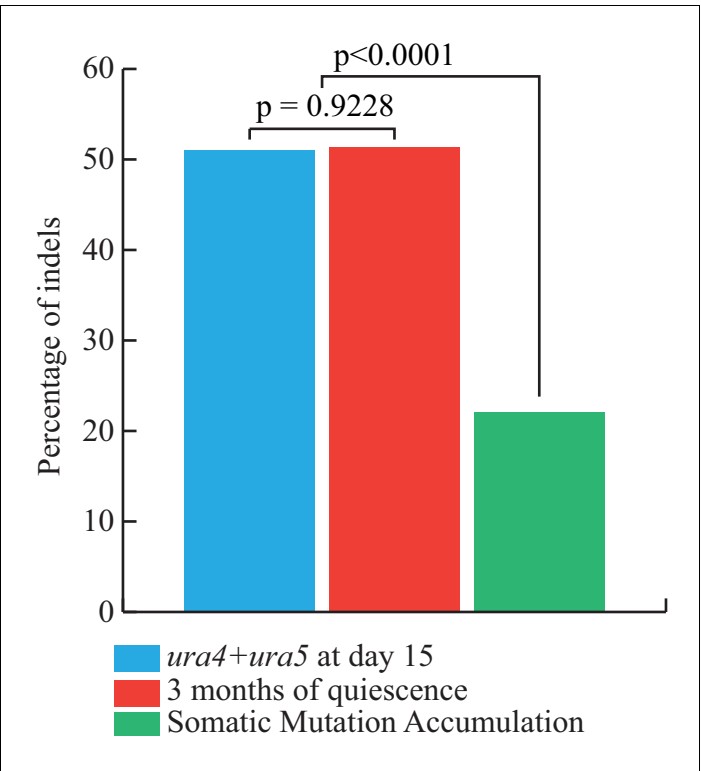

**Figure 6.** Corrected versus observed proportions of Indels. In blue, we report the percentage of Indels expected from the FOA experiments, that is the ratio of the estimated slopes of accumulation of INDELS and SNVs (using method 1). In red, the observed fraction of INDELS after 3 months of quiescence. In green, the observed fraction of INDELs in the MA lines from (*Farlow et al., 2015*) P-values are computed using homogeneity chi2 tests.
DOI: https://doi.org/10.7554/eLife.27469.015

alternatively expose the genome to natural selection that progressively shapes its composition, size and ability. During growth, the universal substitution bias toward AT preference along with the domination of insertions over deletion have been reported in numerous studies (*Hershberg and Petrov, 2010*; *Lynch, 2010*). The fact that the replication-driven mutational bias has not yet reached an equilibrium strongly suggests the existence of forces capable of counterbalancing it. Here we propose that Chronos is this novel mutational force that impact on the genetic material. Separate and alternate modes of mutagenesis and selection allow compensatory mutations to arise in one phase of the life cycle and fuel the phenotypic evolution simultaneously into the subsequent phase, a notion that might impact on germ cells genetics. On the contrary, the super-housekeeping genes

**Table 6.** Phenotypic alterations as a function of time spent in quiescence

| Time in G0 | # of colonies w/phenotype | 18°C | 37°C | KCl 1M | Ca(NO₃)2 0.15M | CaCl₂0.3M | TBZ 15 μg/ml | HU 4 mM | SDS 0.01% |
|---|---|---|---|---|---|---|---|---|---|
| 1 day | 0/384 | - | - | - | - | - | - | - | - |
| 1 month | 0/384 | - | - | - | - | - | - | - | - |
| 2 months | 4/376 | - | - | 1 | 1 | 4 | - | - | - |
| 3 months | 6/334 | 1 | 1 | 1 | 1 | 3 | 3 | 2 | 1 |

(-) indicates the absence of a selected phenotype (all the cells form colonies) The numbers in the columns indicate the number of colonies exhibiting a sensitivity to a given treatment (some mutants display a sensitivity to multiple treatments).

Serial dilutions were spotted on rich medium plates containing (or not) the drugs at the indicated concentrations. (TBZ): Thiabendazole; (HU): Hydroxyurea; (SDS): Sodium Dodecyl Sulfate

DOI: https://doi.org/10.7554/eLife.27469.017

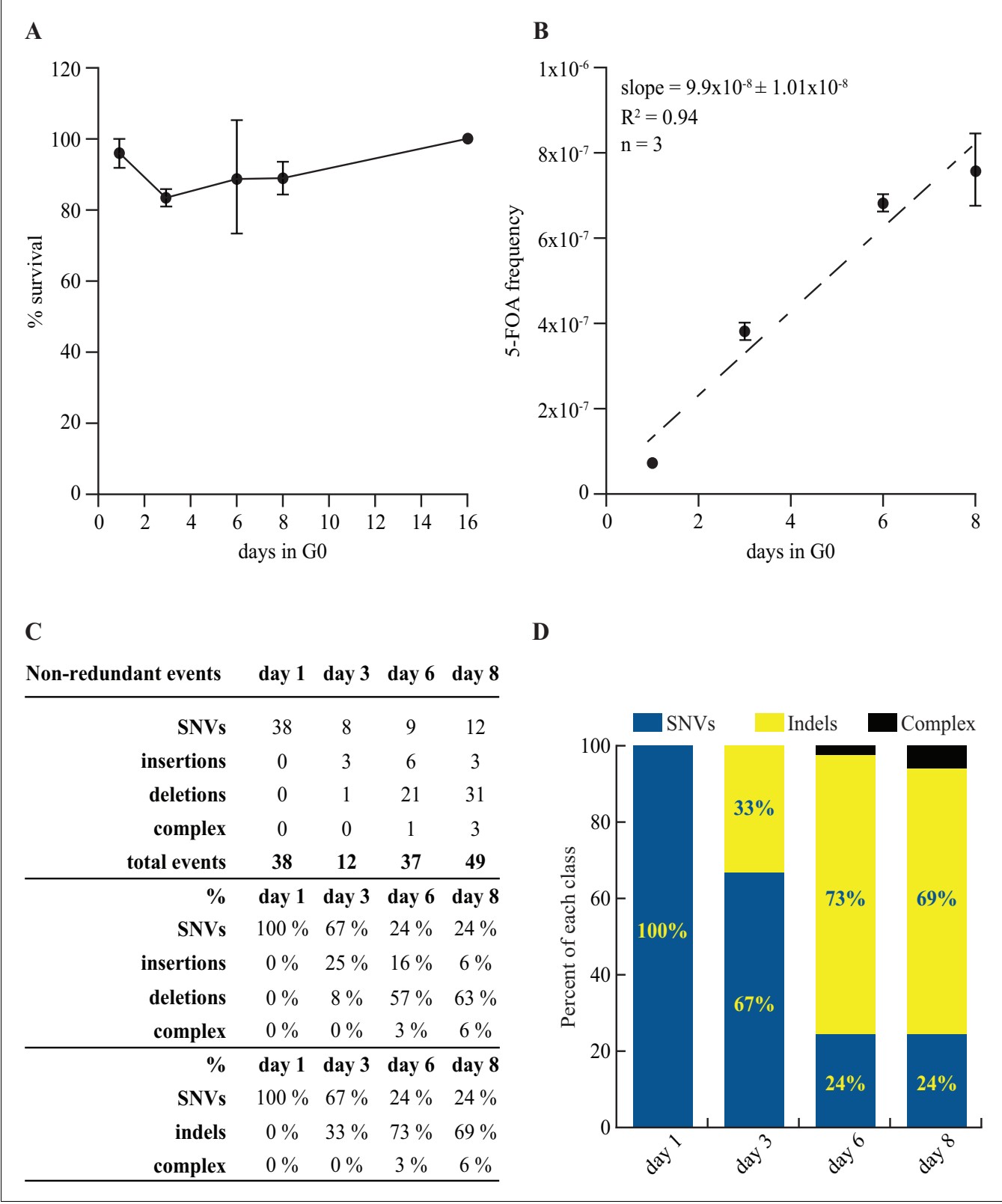

**Figure 7.** Mutation rate and spectrum in pcn1-K164R mutant strain. (**A**) Survival curves in G0 of 3 independent prototrophic pcn1-K164R clones as a function of time. The standard error is indicated. (**B**) Frequency with standard error and slope of FOA^R accumulation in G0 of 3 independent pcn1-K164R clones. The R square value is indicated. (**C**) Table and (**D**) graphical representation of the distribution among SNVs and indels of the *ura4* and *ura5* mutations accumulating over time in pcn1-K164R quiescent cells. The total number of the non-redundant events analyzed is indicated in the table.

*Figure 7 continued*

DOI: https://doi.org/10.7554/eLife.27469.018

(*Yanagida, 2009*) constrain evolution with a strong vector of conservation, since they are required for proliferation, quiescence and the transitions from one to the other (*Williams and Williams, 1957*).

The genetics of quiescence underscores the importance of a time-dependent process (*Goldmann et al., 2016*; *Hazen et al., 2016*; *Kumar and Subramanian, 2002*; *Ségurel et al., 2014*) to the overall mutation spectrum. This time-dependent process should also help to fine-tune the accuracy of the 'molecular clock' that measures, in units of time (*dos Reis et al., 2016*), the evolutionary distance of two closely related species with their common ancestor. Such hypotheses are accessible to experimental and modeling approaches and are of great interest for evolutionary, developmental and human-health perspectives.

## Materials and methods

### Strain and sanger sequencing of ura4 and ura5 mutants

The stable prototrophic M-smt0 PB1623 strain was used in all our experiments. We identified the FOA$^R$ mutant strains as *ura4$^-$* and *ura5$^-$* mutants by genetic crosses with a known *ura4Δ* strain, purified their DNA, PCR amplified the gene of interest, analyzed the PCR fragment by agarose gel electrophoresis and subjected it to Sanger sequencing.

### Library construction and sequencing

Library construction and sequencing was performed by Illumina HiSeq 2500 following the manufacturer's instruction. Base calling was performed using CASAVA 1.9. For each strain, one insert size (ranging from 400 to 800 bp) library was constructed and sequenced. After initial quality control assessment with FastQC version 0.10.1 (*Chen et al., 2012*) fqCleaner (l = 80; q = 30) was used to trim the tails of the reads if the Phred quality dropped below 30.

### Alignment-based assembly

We sequenced 12 strains per lane on an Illumina HiSeq 2500, aligned the resulting reads to the *Schizosaccharomyces_ pombe*.ASM294v2.23 DNA reference genome with BWA-MEM; version 0.7.5a. (*Li and Durbin, 2009*). SAMtools version 0.1.19 and Picard version 1.96 (http://picard. sourceforge.net) were used to process the alignment files and to mark duplicate reads. The coverage in our experiments ranges from 60 to 120, with an average of 80. SNVs and small indels were called using GATK version 2.7–2 (*McKenna et al., 2010*). We applied quality score recalibration, indel realignment, duplicate removal, and performed SNV and INDEL discovery and genotyping using standard filtering parameters or variant quality score recalibration according to GATK Best Practices recommendations (*DePristo et al., 2011*; *Van der Auwera et al., 2013*). In addition to the GATK analysis, we combined three programs, Pindel (*Ye et al., 2009*), Prism (*Jiang et al., 2012*), and SOAPindel (*Li et al., 2013*) dedicated to the detection of INDELs to search for additional variants not detected by GATK. Only variants detected at least 10-times in a sample and not found in any other strain sequenced from the same G0 pool were considered.

### Filtering

For GATK, Pindel, Prism and SOAPindel analyses, only variants detected at least 10-times in a sample and not found in any other strain sequenced from the same G0 pool were considered. For GATK and indels detection, we have determined the FDR by Sanger sequencing on a larger set of previously sequenced strains. 20 random SNVs whose quality scores ranged from 21 to 1700 were analyzed. Only the lowest score (21; one occurrence) turned out to be a false positive, yielding an FDR of 0.05. Concerning the indels, all the variants detected by at least two approaches, including GATK, turned out to be true by Sanger sequencing. For the variants only detected by Prism, we sequenced 13 occurrences and found that only the two lowest scores (DP10) were false positives (FDR = 0.15).

Pindel yielded the poorest yield of variants that were systematically true and called by at least one other program. SOAPindel called a large number of variants that were dispatched into five classes according to their type of output in the VCF file. We kept only the relevant calls labeled as HP = A_B or HP = X_N in the vcf file for which the FDR determined on 14 variants was 0.2143 (in the three other classes, 54 occurrences, only five were true and were not taken into account).

## Miscellaneous

Template DNA fragments were hybridized to the surface of paired-end (PE) flow cells (HiSeq 2500 sequencing instruments) and amplified to form clusters using the Illumina cBotTM. Paired-end libraries were sequenced using $2 \times 120$ cycles of incorporation and imaging with Illumina SBS kits. For the HiSeq 2500, $2 \times 101$ cycles with SBS kits v3 were employed. Each library was initially run, assessing optimal cluster densities, insert size, duplication rates and comparison to chip genotyping data. Following validation, the desired sequencing depth (>60X) was then obtained. Real-time analysis involved conversion of image data to base-calling in real-time.

## Estimation of the total mutation rate

The experimental assay based on 5-FOA resistance can be used to estimate the fraction of mutations that give rise to a phenotype. All the mutations that we have observed are single mutation events that invalidate either the $ura4^+$ or $ura5^+$ genes, hence resulting in the FOA$^R$ phenotype. We now want to infer the total mutation rate, both for SNVs and indels independently. We hypothesize that most indels will be deleterious to the genes and that therefore the total indel rate is close the observed indel rate that results in FOA$^R$ (for genes - 50% of the genome). On the contrary, many SNVs are likely to exhibit no phenotype and, therefore, we observe only a fraction of the total number of SNVs.

## The gene model

As we only observe mutations among SNVs that lead to a phenotype, we have estimated the fraction $f$ of non-synonymous SNVs that lead to a phenotype using three alternative methods. All three methods exploit different observations and lead to different estimators. As none seems obviously better than the two others, we report the three results in *Table 4*. In all three methods, the underlying gene model is the same and is indicated in *Figure 8*.

## Method 1 – From the Aa saturation

The first method assumes that the non-synonymous SNVs that lead to a phenotype are evenly distributed among an unknown number 'essential' amino-acid (Aa). Whenever an essential Aa is mutated, there is a FOA$^R$ phenotype. We thus simply compute the observed distribution of the number of different mutations per Aa (from *Table 4* -raw), from which we estimate the total number of essential Aa using maximum likelihood (see below). It thus assumes no mutational bias and uses the distribution of *different* mutations per Aa.

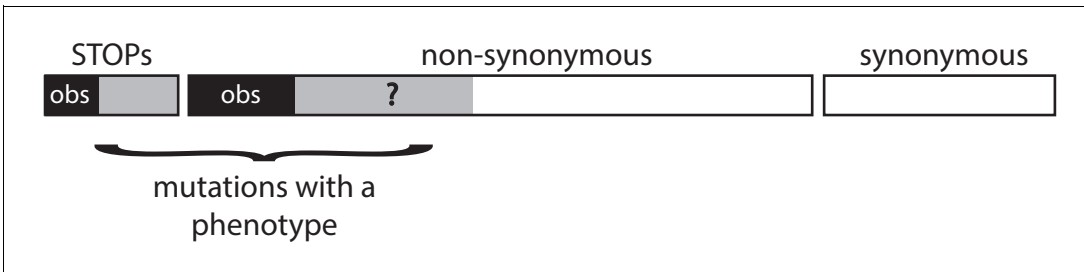

**Figure 8.** Schematic representation of a gene. In all three methods, the gene model is the same. We observe only a subset of all possible SNVs (the back areas) among all SNVs that lead to a phenotype (the black and grey areas). All synonymous SNVs do not exhibit a phenotype. The methods aim at estimating the unknown proportion f of the non-synonymous SNVs that leads to a phenotype.
DOI: https://doi.org/10.7554/eLife.27469.019

## Method 2 – From the nucleotide saturation

The second method assumes that non-synonymous SNVs that lead to a phenotype can be targeted several times by independent mutations. Whenever one of them is mutated, there is a phenotype. We thus build the distribution of the number of independent recurrent mutations for each non-synonymous SNVs (from *Table 4* -filtered), which is then used to estimate the total number of non-synonymous SNVs that lead to a phenotype using maximum likelihood (see below). It thus assumes no mutational bias and uses the distribution of *independent identical* mutations per non-synonymous SNV.

## Method 3 – *Lang and Murray (2008)*

The last method, proposed by *Lang and Murray (2008)*, equates the level of saturation of STOPs and non-synonymous mutations that lead to a phenotype. The level of saturation of STOPs is estimated by the fraction of STOPs that are reported (over the total number of possible STOPs). It thus assumes no mutational bias and uses the observed number of STOPs and non-synonymous SNVs.

## Estimating the total number of targets from *k*-distribution by maximum likelihood

The model assumes that a set of *k* targets give rise to a phenotype once mutated (targets are either essential Aa –method 1– or non-synonymous mutations with a phenotype –method 2–). All *m* mutations are uniformly distributed among the targets. Once all mutations are assigned, a target can have 0, 1 or more mutations. Only the number of non-mutated targets (0 hits) is unknown and therefore to be estimated. We estimated it by maximum likelihood using the distribution of mutations per target, hereafter the *k*-distribution.

We observe a total of *m* mutations uniformly distributed among the *k* targets, where target *i* has $m_i$ mutations:

$$m = \sum_{i=2}^{k} m_i$$

As each mutation has a probability 1/*k* to occur at a particular target, the set of $m_i$ is given by the multinomial probability distribution:

$$\frac{m!}{\prod_{i=1}^{k} m_i!} \times \frac{1}{k^m}$$

We consider in our model that all targets are exchangeable. Defining $k_j$, as the number of targets that have *j* substitutions, the number of exchangeable configurations is given by:

$$\frac{k!}{\prod_{i=0}^{max} k_i!}$$

, where *max* is the highest observed number of mutation for a target. Therefore, the product of the two previous terms once rearranged, is the likelihood of the overall observed distribution of ($k_1$, $k_2$, … $k_{max}$) given *k*:

$$P(k_1, k_2, ..., k_{max}|k) = \frac{m!k!}{\prod_{i=0}^{max} k_i!(i!)^{k_i}} \times \frac{1}{k^m}$$

with

$$m = \sum_{i=1}^{max} i k_i$$

and

$$k_0 = k - \sum_{i=1}^{max} k_i$$

We thus computed numerically the maximum likelihood estimate of k as well as its associated 95% credibility interval (Table S3).

## How many SNVs can lead to a phenotype?

As we observe only SNVs that disrupt the function of the genes, we assumed that all synonymous substitutions have no phenotype, that all STOP substitutions do, as well as a fraction $f$ of the non-synonymous SNVs. Mathematically, it is expressed as $[P_{STOP} + f\,P_{NS}]$, where $P_{STOP}$ $P_{NS}$ are the proportions of STOP and Non-Synonymous mutations among all possible mutations.

## How many indels can lead to a phenotype?

We further assumed that most insertions or deletions have phenotypes, except when the reading frame is kept intact. For these *in-frame* indels (which size is a multiple of 3), there is a phenotype only when an essential Aa (estimated by $f$) is deleted or disrupted by an insertion.

We retrieve the size distribution of indels and inflated the 3 bp indels (0.053 for URA4 and 0.019 for URA5) by a factor of $1/f$ and the 6bp indels (0.019 for URA4 and 0.0 for URA5) by a factor of $1/(1-(1\;f)^2)$. Larger insertions/deletions are all assumed to have a phenotype. Mathematically, it is expressed as $1/[1 + P_{3nt} / f + P_{6nt} / (1-(1\;f)^2)]$, where $P_{3nt}$ $P_{6nt}$ are the proportions of indels of size 3nt or 6nt among all observed indels (*Table 4—source data 1*), in which the values for f were the ones estimated from *Table 4* for each category. The estimated mutation rates are reported in *Table 4—source data 2*.

## Acknowledgements

We thank Karl Ekwall and Stéphane Marcand for their critical reading of the manuscript. This work was supported by the grants ANR-13-BSV8-0018 and ANR-12-BSV7-0012 Demochips from the Agence Nationale de la Recherche (France) to BA and GA, respectively.

## Additional information

### Funding

| Funder | Author |
| --- | --- |
| Agence Nationale de la Recherche | Guillaume Achaz<br>Benoit Arcangioli |

The funders had no role in study design, data collection and interpretation, or the decision to submit the work for publication.

### Author contributions

Serge Gangloff, Resources, Data curation, Software, Formal analysis, Supervision, Validation, Investigation, Methodology, Writing—review and editing; Guillaume Achaz, Conceptualization, Resources, Data curation, Software, Formal analysis, Supervision, Validation, Investigation, Methodology, Writing—original draft; Stefania Francesconi, Investigation, Methodology; Adrien Villain, Resources, Software, Formal analysis, Investigation, Methodology; Samia Miled, Formal analysis, Investigation; Claire Denis, Investigation; Benoit Arcangioli, Conceptualization, Formal analysis, Supervision, Funding acquisition, Investigation, Methodology, Writing—original draft, Project administration, Writing—review and editing

### Author ORCIDs

Serge Gangloff https://orcid.org/0000-0003-1333-6091
Benoit Arcangioli https://orcid.org/0000-0002-1371-1405

Decision letter and Author response
Decision letter https://doi.org/10.7554/eLife.27469.026
Author response https://doi.org/10.7554/eLife.27469.027

# Additional files

### Supplementary files
• Supplementary file 1. Position and nature of the alterations yielding 5-FOA resistant colonies in *ura4*[+] and *ura5*[+] genes.
DOI: https://doi.org/10.7554/eLife.27469.020

• Supplementary file 2. Mutations detected in the 243 colonies sequenced after 3 months in G0.
DOI: https://doi.org/10.7554/eLife.27469.021

• Transparent reporting form
DOI: https://doi.org/10.7554/eLife.27469.022

### Major datasets
The following dataset was generated:

| Author(s) | Year | Dataset title | Dataset URL | Database, license, and accessibility information |
| --- | --- | --- | --- | --- |
| Gangloff S | 2017 | Quiescence pombe sequencing | http://www.ncbi.nlm.nih.gov/bioproject/413662 | Publicly available at the NCBI BioProject database (accession no: PRJNA413662) |

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
