## [Decision Letter]

Thank you for submitting your article "Quiescence unveils a novel mutational force in fission yeast" for consideration by *eLife*. Your article has been reviewed by three peer reviewers, and the evaluation has been overseen by Naama Barkai as the Senior and Reviewing Editor. The following individuals involved in review of your submission have agreed to reveal their identity: Nick Rhind (Reviewer #1) and Primo Schär (Reviewer #3).

Summary:

The study shows that starved fission yeast cells still mutate their genome at high rate, and define the associated mutation spectrum. All three reviewers found this result interesting, but raised substantial concerns that will need to be addressed using more experiments and analysis.

Essential revisions:

There are four main issues that the reviewers are only partially convinced on and that needs to be clarified:

1) Lack of cell replication: please validate quiescence by providing quantitative data on the population homogeneity with regard to cell cycle and proliferation.

2) The mechanism generating the mutations: Please test the hypothesis/rule out the possibility that the translesion polymerases as the basis of the new spectrum.

3) Please also examine the possibility that nucleotide imbalance (caused by FOA or metabolic shift) triggers repair-mediated mutational events. An insight into this question may be gained by comparing mutation spectra arising under FOA (ur4/5 sequences) and those occurring spontaneously (genome sequencing) in a quantitative manner.

4) Please extend the analysis of sequencing data as proposed by the reviewers.

*Reviewer #1:*

Gangloff et al. present a simple but convincing demonstration of the difference between the mutation spectrum of log phase fission yeast and those in a quiescent population. The fact that the quiescent mutation rate is so high is surprising – my intuition was that it should be low in a metabolically less active state presumably designed for durability – and the contrast between the two spectra raise interesting issues of mutational homeostasis. Nonetheless, it seems to me that this observation is more intriguing than informative. To have a significant impact in the field, I would think that one would want to provide some insight into the mechanisms by which these mutations accumulate or a more rigorous formal treatment of how the two spectra balance and what that says about the amount of time fission yeast spend growing versus quiescent.

In addition, attention to the following issues would strengthen the paper.

It is not clear how useful it was to name the quiescence mutational spectrum "Chronos", especially since Chronos does not appear again in the paper.

Although it is clear that the populations analyzed are quiescent, it is harder to say much specific about the cell leading to the selected mutations. In particular, calling these mutations replication independent is probably too strong. It would be useful to include a more explicit discussion of whether they may be rare replicating cells and, if so, what the different mutation spectrum might say about replication in quiescent populations.

The speculation about the mutational spectra in mammalian gametes is far removed from the results presented in the paper and does not add much insight to either the fission yeast or the mammalian biology. It should be saved for an opinion piece rather than a research paper.

Speculation, or preferably data, that would be more useful would address the mechanism of the mutations in quiescent populations.

*Reviewer #2:*

This is an interesting paper that studies changes in the spectrum of mutations in *S. pombe* as cells are held in starvation medium, lacking nitrogen. The frequency of mutations increases linearly with time over two weeks but apparently increases at 3 months. The spectrum of mutation changes, with somewhat more deletions than insertions (though this is not evident in the whole genome survey of 123 strains after 3 months).

There are a number of statements that require clarifications.

1) Introduction, second paragraph. According to Marguerat et al. 2012, "These cells remain metabolically active by recycling nitrogen, become highly resistant to multiple stresses, and survive for months (Yanagida, 2009)." This raises the fundamental question whether quiescent cells are really devoid of DNA replication.

Would the authors know if 0.1% were able to replicate their DNA, with a different mutational profile? Could they show whether EdU or BrdU are incorporated, and if so, are these incorporations over long segments or only in repair-sized patches?

If additional experiments are suggested, they need to be justified by the reviewer. I am not convinced that metabolically active cells, able to recycle nutrients, are truly non-replicating. It seems equally likely that a subset of cells ("persisters") might indeed fully replicate their genomes. A more direct way of showing the extent of new DNA synthesis is highly desirable, but other reviewers may disagree.

2) Results, second paragraph. It is curious that a 50-50 mixture of ura4 and WT cells survives better than either population alone. This appears to be statistically significant. Is it? In any case the issue is not whether ura4 survives less, but whether it survives better (which would suggest some possibility of selection). When the 50-50 mixture survives, is there a change in the proportion of ura4 cells?

3) Results, sixth paragraph. What happens to mutations when the medium is not changed every two weeks? And what does it mean that the medium is changed to "maintain oligo elements" (are these nucleotides?).

4) "Mutations found more than once in clonal population" – do the authors mean "more than once in different clonal populations"?

5) Figure 1. Most mutations are apparently +/-1 indels. Do they arise in homonucleotide runs? I am not sure what "low complexity" sequences means.

6) Results, fourth paragraph. I find statements about evolutionary pressures overstated, based on this small survey of two genes whose nucleotide sequence may – or may not – reflect the entire genome. The small changes in + and – indels are insufficient for such a claim, especially because the 3-month data don't support the claim.

In this regard, is this whole genome or exome sequencing? If whole genome, were there no changes in copy number or other changes in the repeated sequences around centromeres, rDNA and elsewhere? These changes would support the idea that nutritional restriction would favor deletion to make a smaller, easier-to-replicate genome, even if under supposedly nonreplicating conditions.

7) Figure 2 I am unable to evaluate the corrections to mutation frequencies based on the author's new method of estimating "essential" amino acids. I do not see how they evaluated the better predictive fit of this method relative to Lang and Murray's, which seems to be a straightforward calculation based on the idea that STOP codons become over-represented because many nonsynonymous changes are still functional. Unless the GC content is 50%in coding regions, and given that some SNVs are much more frequent than others, the statement that each mutation has a probability of 1/k may not be correct. In any case please provide the criteria by which the new method outperforms Lang and Murray and better explain the corrected frequencies.

8)Results, seventh paragraph. What was the estimated mutation frequency for 3 months?

9) Table 3. Is there a way to show which of the 4 or 6 mutations has which of the 8 phenotypes? And, since there was only about 0.7 mutations per genome, if I read correctly, it should be evident – or highly suggestive – what mutation in what gene caused these phenotypes. Do these mutations make sense in terms of their phenotypes?

10) The biggest question surrounding this approach is *why* these changes in mutation occur. In bacteria, SLAM reflects the increased use of mutagenic DNA polymerase DinB, leading to a change in mutational spectrum. How do bacterial mutation rates change with time? In budding yeast, DNA repair has an 1000-times increase in mutation rate compared to replication, with evidence of the use of a different DNA polymerase in some instances. None of these studies is cited, although Foster's early work on adaptive mutation is. Rosenberg's extensive later work deserves attention. Ditto Strathern's and Haber's yeast work.

So, if starving (quiescent) cells are subject to the use of a different polymerase or an altered mechanism of DNA repair, there well could be a change in mutation rate. These possibilities should be explored, perhaps instead of rather unconnected musings on the differences between sperm and ovum.

*Reviewer #3:*

The manuscript by Serge Gangloff and co-workers presents data on mutagenesis in quiescent *S. pombe* cells. The authors combined conventional and next generation sequencing to identify genetic alterations that occur in *S. pombe* cells held in a non-replicating state by nitrogen starvation for up to 3 months. In a first approach they analyse in a time course of 15 days single nucleotide changes and insertion/deletion mutations in the ura4 and ura5 genes in 5-FOA resistant cells. This dataset shows that, following entry into the quiescent state, mutations accumulate in a time-dependent manner, showing a characteristic spectrum that is distinct from replication-associate mutagenesis. A second approach identifies genomic mutations in cell populations following 3 months of nitrogen starvation and essentially corroborates the replication-independent mutational features. The authors conclude that replication- and quiescence-associated mutagenesis represent two alternative mutation drivers, facilitating a balanced evolution of genomes.

This work addresses a long-recognized but difficult to study, yet important gap of knowledge in mutagenesis, the highly complex question of the stability of genomes in non-replicating cells. The authors approach is based on the well-defined fission yeast model, where nitrogen-starvation can induce a "physiological" quiescent state, as well as of state of the art targeted and genome-wide analyses. The experimental setup has generated highly interesting data that one would like to explore further, both conceptually and mechanistically. While I greatly appreciate the authors efforts to advance this area of research and see the potential importance of the work, my impression is that the state of the current manuscript is somewhat immature; the data is purely descriptive at this point and, rather than using them to develop (and test?) hypotheses regarding the molecular mechanisms promoting mutagenesis in quiescence, the authors focus on genome evolutionary interpretations that seem to stretch the evidence presented considerably. These are definitely interesting considerations but the experimental support is not compelling. In my opinion the manuscript would benefit greatly by the addition of some controls (G0-phase validation), more rigorous analyses and discussion of the actual data, and a more thorough consideration/discussion (perhaps testing) of the potential mechanisms involved in quiescence mutagenesis. Addressing the following questions might help.

1) Mutational baseline: I have a number of questions relating to the discrimination of replicating from quiescent cells, i.e. the definition of baseline, which is critical here. (i) It is assumed that a majority of cells enter G0 under nitrogen-starved conditions. While this it probably correct, was the cell cycle status validated in these experiments and where proportions of G0 cells assessed? (i) Vice versa, what would be the proportion of quiescent cells in proliferating populations of *S. pombe* cells? (iii) How are pre-existing persisters dealt with and excluded? (iv) Why isn't there more fluctuation in mutation frequencies at end of replication, early quiescence? How many cultures were tested?

2) FOA mutation assay and time of occurrence of mutations: The mutation assay described bases on the selection of 5-FOA resistant survivors. FOA itself perturbs the uracil metabolism and, thereby also the dNTP balance. This in itself can generate a shift in mutational spectra. In the experimental setup used, mutations may emerge only after re-plating of cells on FOA media, possibly by repair of replication of DNA damage that was induced during quiescence. Nucleotide imbalance may then affect the type of mutation generated. Was this considered? What is the authors’ take on this? I think, this should be addressed briefly in the Discussion.

3) The FOA-based mutation assay is not fully unbiased as it scores functional changes related to uracil mechanism only. This limitation should be phrased in the description of the assay. To what extent could the spectra shift be explained by the FOA effect on nucleotide metabolisms (changes involving T?).

4) Mutation spectra: I suggest displaying the frequencies of individual single-nucleotide changes (e.g. Figure 1, respective data in Table 1 and Table 2) as normalized to the total number of relevant scorable base pairs. This would provide mora accurate information about true mutation preferences.

5) CpG mutagenesis: It is interesting, very interesting indeed, that CpG dinucleotides are preferential sites of mutation. Again, it would be interesting to see if and to what extent the normalized frequency (normalized to total number of such sites in the scorable sequences) is higher than predictable (statistics against all dinucleotides). This is likely, given the coding importance of these nucleotides and the fact that they not only preferentially mutate by C (or 5mC) deamination but also by alkylation of G (6-methylguanine), for instance.

6) Statistics in general: Are the differences in spectra between day 1 and day 8+15 statistically significant (Figure 1 and Figure 2)?

7) Normalizations: I would be interested to have the analysis performed in Figure 2 extended for individual SNVs (again normalized to the total number of scorable bases in each category).

8) Genome-scale mutation assessment: This precious dataset should, in my opinion, be explored further, for instance to validate the CpG bias, or sequence context biases, or the FOA bias. I could imagine there is quite some potential there. Also, it seem important to me to compare this dataset to one of exponentially growing cells (same number of cell after similar time of continuous cultivation). In my opinion, this experiment is a real strength of the paper.

9) Discussion: The Discussion is interesting and stimulating but rather far reaching. Given that some caveats exist (FOA, persisters…) and the datasets are not exhaustively explored and also entirely descriptive, I would prefer to have some discussion of the actual data, their strengths and weaknesses, and the mechanisms of mutagenesis implicated. What could be the molecular reason for the spectrum shift?

[Editors' note: further revisions were requested prior to acceptance, as described below.]

Thank you for resubmitting your work entitled "Quiescence unveils a novel mutational force in fission yeast" for further consideration at *eLife*. Your revised article has been favorably evaluated by Naama Barkai as the Senior and Reviewing Editor and two reviewers.

The manuscript has been improved but there are some remaining issues that need to be addressed before acceptance, as outlined below:

Please acknowledge that repair processes could indeed give 1000x increases that you see.

*Reviewer #1:*

I raised two main points about the Gangloff manuscript: the question of whether a minority of cells may be replicating during quiescence and the opinion that, without some mechanistic insight into the cause of the mutations during quiescence or a more rigorous formal treatment of the evolutionary implications of the observation, the work was more intriguing than informative.

The authors addressed my first point, although somewhat indirectly. Instead of measuring the number of cells replicating, they measure the number of cells dividing and conclude that since very few cell divide, very few could be replicating. The other possibility is that cell occasionally replicate, but then arrest in a G2 G0, which has been reported in fission yeast. Nonetheless, I am persuaded by their data that there is unlikely to be a significant fraction of replicating cells and that their mutations are therefore most likely replication independent.

The authors’ response to the second point is less satisfying. They show that TLS is not a significant contributor to the mutation spectrum they observe, but offer no mechanistic insight beyond that and still speculate about the evolutionary significance of their result in only vague and qualitative terms. So, while I agree that this intriguing observation will cause people to think differently about DNA stability in quiescence, it does not provide much guidance as to what the new way of thinking about it should be.

*Reviewer #2:*

The authors have generally made a very good effort to address the reviewers' concerns. Showing that a PCNA mutation did not alter significantly the mutation rate was important (though it remains then a question what is the source of the elevated mutations, if they are not replicating or subject to the use of translesion polymerases to deal with damage in nondividing cells).

I am still stuck on one point, which I would ask the authors to address directly, based on their own statement in the rebuttal: "If these replicating cells were generating the FOAR mutants observed during quiescence, we will have to conclude that they generate 1,000 times more mutations per replication." That was my point originally – the papers now cited by Dr. Arcangioli from yeast DNA damage studies make exactly that argument: repair DNA synthesis is 1000x more mutagenic than normal replicative errors. So they should acknowledge that – although they believe their data make this unlikely – the very high rate of errors in DSB repair could indeed give the numbers that their CHRONOS study reveals.

---

## [Author Response]

Essential revisions:There are four main issues that the reviewers are only partially convinced on and that needs to be clarified:1) Lack of cell replication: please validate quiescence by providing quantitative data on the population homogeneity with regard to cell cycle and proliferation.

In the Results section, we have introduced an additional analysis of the cell population homogeneity in quiescence with Table 1 and added:

“To determine the homogeneity of the cell population during quiescence, we analyzed the size and the proportion of cells exhibiting a septum during 15 days (Table 1). […] This result indicates that the proportion of cells dividing or replicating during quiescence is very low.”

See also the responses to reviewer comments.

Early in the Discussion section, we added:

“Extensive work has shown that quiescence is a well-controlled and conserved process {Yanagida, 2009; Lee et al., 1988; Petersen and Hagan 2005}. […] Therefore, we favor the Chronos model where DNA lesions that are arising during quiescence {Mochida and Yanagida, 2006; Ben Hassine and Arcangioli, 2009} are repaired with errors that generate mutations with time.”

“Initially, Luria and Delbruck {Luria and Delbrück, 1943} expressed the mutation rate *per unit time*, that was transformed to *per generation* later by Lea and Coulson {Lea and Coulson, 1949}. […] Our work proposes that quiescence is a genetic active state that participates to the mutation fraction that is expressed in unit of time.”

2) The mechanism generating the mutations: Please test the hypothesis/rule out the possibility that the translesion polymerases as the basis of the new spectrum.

PCNA lysine 164 ubiquitination regulates TLS by promoting switching from replicative to specialized polymerases that are able to bypass different DNA lesions. To address the possibility that TLS is involved in the quiescence mutational signature we have analyzed the viability, the mutation accumulation and the spectrum of mutants arising in the translesion DNA synthesis mutant pcn1-K164R.

We found that the pcn1-K164R mutant, that cannot switch from replicative to TLS mode (i) maintains high viability during quiescence, (ii) accumulates almost twice FOA^R^ mutants compared to the wild type strain (iii) with more indels than SNV mutations and a domination of deletions over insertions.

We added one figure (Figure 7) and the following into the Discussion section:

“If the mutations are not generated by errors during DNA replication, what are the mechanism(s) underlying Chronos? […] Further investigations of the DNA repair pathways at work during quiescence are needed not only to provide information on the importance of each DNA repair pathway but also valuable insights into physiological source of the spontaneous DNA lesions in quiescence.”

3) Please also examine the possibility that nucleotide imbalance (caused by FOA or metabolic shift) triggers repair-mediated mutational events. An insight into this question may be gained by comparing mutation spectra arising under FOA (ur4/5 sequences) and those occurring spontaneously (genome sequencing) in a quantitative manner.

To assess the potential impact of FOA on the mutations accumulation and spectrum we added a paragraph in the Results section, one in Materials and methods and one Table (Table 3).

“A possible caveat of our study is that FOA is known to perturb the metabolism of uracil and thymidine and might thus impact differently the mutational spectra of growing and quiescent cells. […] The 16 unique mutations suggest a frequent heterozygosity of the mutations that can be revealed only after cell division.”

Please see also the response to reviewer 3.

4) Please extend the analysis of sequencing data as proposed by the reviewers.

The whole genomes of 243 strains isolated after three months of quiescence are now reported. We updated the related section. The mutations accumulating during quiescence either from the estimated mutation rates from the FOA experiments or from the three months old quiescent cells are almost identical and different from the mutation accumulation results provided by Lynch et al. (Figure 6).

Therefore, we clarified the Chronos profile definition as a genetic force generating similar proportion of SNVs and Indels together with an enrichment of deletions. We also added information on the CpG dinucleotides (Figure 2—figure supplement 2)

We added:

“Altogether, quiescence in fission yeast reveals Chronos as a new genetic force generating similar proportions of SNVs and Indels, together with an enrichment of deletions.”

We also added Figure 5 and Figure 6 showing in a quantitative manner (point 3) the Poisson distribution of the mutations per genome and the similarities and differences between the growing and quiescent mutational profiles.

We added in the Results section and figure legends:

“We also analyzed the distribution of the number of mutations (143) per genome (237) that fits perfectly a Poisson distribution. […] This result shows that a unique mutational force, equally affecting all genomes, is at work during quiescence.”

Figure 5: Distribution of mutations among the 237 genomes sequenced. […] The goodness of fit is excellent (Pchi2=0.56).

Figure 6: Corrected versus observed proportions of Indels

In blue, we report the percentage of Indels expected from the FOA experiments, that is the ratios of the estimated slopes of accumulation of INDELS and SNVs (using method 1). In red, the observed fraction of INDELS after 3 month of quiescence. In green, the observed fraction of INDELs in the MA lines from {Farlow et al., 2015}. P-values are computed using homogenity chi2 tests.

Figure 2—figure supplement 2: Normalized percentage of each dinucleotide in the open reading frames of *ura4*^+^ and *ura5*^+^after one day in quiescence. The percentage of every dinucleotide mutated at day one is compared to the frequency of occurrence of the same dinucleotide in the respective open reading frame.

We also sequenced 6 genomes of the three-months aged cells exhibiting phenotypes and identified the putative causative mutations and genes of interest. We added at the end of the Results section (reviewer 2):

“We sequenced the genomes of the six strains isolated after three months and found mutations mapping in *alg5, alg8, aur1, ccc2, gdi1* and *oca8* genes. These genes encode for cell wall and vacuole/membrane trafficking functions that might improve viability during quiescence.”

Reviewer #1:Gangloff et al. present a simple but convincing demonstration of the difference between the mutation spectrum of log phase fission yeast and those in a quiescent population. The fact that the quiescent mutation rate is so high is surprising – my intuition was that it should be low in a metabolically less active state presumably designed for durability – and the contrast between the two spectra raise interesting issues of mutational homeostasis. Nonetheless, it seems to me that this observation is more intriguing than informative. To have a significant impact in the field, I would think that one would want to provide some insight into the mechanisms by which these mutations accumulate or a more rigorous formal treatment of how the two spectra balance and what that says about the amount of time fission yeast spend growing versus quiescent.

We agree that knowing the DNA repair pathways and mechanisms operating during vegetative and quiescence states are equally important, but addressing experimentally the major DNA repair pathways at work during quiescence requires an effort well beyond the scope of the present work. Nonetheless, as requested, we added a new set of data on the role of the TLS pathway with the conclusion that its impact is rather weak during quiescence (Essential revisions 2).

In addition, attention to the following issues would strengthen the paper.It is not clear how useful it was to name the quiescence mutational spectrum "Chronos", especially since Chronos does not appear again in the paper.Although it is clear that the populations analyzed are quiescent, it is harder to say much specific about the cell leading to the selected mutations. In particular, calling these mutations replication independent is probably too strong. It would be useful to include a more explicit discussion of whether they may be rare replicating cells and, if so, what the different mutation spectrum might say about replication in quiescent populations.The speculation about the mutational spectra in mammalian gametes is far removed from the results presented in the paper and does not add much insight to either the fission yeast or the mammalian biology. It should be saved for an opinion piece rather than a research paper.Speculation, or preferably data, that would be more useful would address the mechanism of the mutations in quiescent populations.

We added new data on the cell population homogeneity during quiescence and discussed the consequences of potential persisters that are not responding to the absence of nitrogen and may potentially replicate during the quiescence period. We also added a new data showing that *ura4* and *ura5* mutations arising during quiescence or growth are similarly impacted by FOA. We removed most of the speculation on mammals and discussed Chronos as a potential vector for time-dependent mutations. (Essential revisions 1 and 3).

Reviewer #2:[…] There are a number of statements that require clarifications.1) Introduction, second paragraph. According to Marguerat et al. 2012, "These cells remain metabolically active by recycling nitrogen, become highly resistant to multiple stresses, and survive for months (Yanagida, 2009)." This raises the fundamental question whether quiescent cells are really devoid of DNA replication.Would the authors know if 0.1% were able to replicate their DNA, with a different mutational profile? Could they show whether EdU or BrdU are incorporated, and if so, are these incorporations over long segments or only in repair-sized patches?If additional experiments are suggested, they need to be justified by the reviewer. I am not convinced that metabolically active cells, able to recycle nutrients, are truly non-replicating. It seems equally likely that a subset of cells ("persisters") might indeed fully replicate their genomes. A more direct way of showing the extent of new DNA synthesis is highly desirable, but other reviewers may disagree.

We agree with all the concerns. We added an analysis of the size and proportion of cells containing a septum during quiescence and discussed the consequence of rare dividing cells in quiescence. (See Essential revisions 1). This experiment indicates that less than one cell every 1000 cells might replicated or divided in quiescence, a detection threshold that could be improved by incorporation of thymidine analogs. However, incorporation of thymidine analogs required the two ectopic genes, thymidine kinase and nucleotide transporter, and complemented minimum liquid media. In addition, BrdU, CldU and EdU are known mutagens that may introduce a bias in the analysis. These compounds are base analogs that will introduce nitrogen to the medium and interfere with quiescence. We also analyzed the distribution of the mutations per genomes after three months of quiescence and found a perfect fit with Poisson distribution (supplemental curve) strongly suggesting that a single mutational force is at work during quiescence.

2) Results, second paragraph. It is curious that a 50-50 mixture of ura4 and WT cells survives better than either population alone. This appears to be statistically significant. Is it? In any case the issue is not whether ura4 survives less, but whether it survives better (which would suggest some possibility of selection). When the 50-50 mixture survives, is there a change in the proportion of ura4 cells?

Figure 1 is showing several experiments performed comparing wild-type, ura4D18 and co-culture. Altogether they are showing that ura4D18 has no advantage compared to wild-type strain or mixed populations and the proportion of uracil auxotrophs is maintained constant for the two weeks (we added a sentence in the Figure 1 legend).

“*ura4D18* has no advantage compared to wild-type strain or mixed populations and the proportion of uracil auxotrophs is maintained constant for the two weeks.”

3) Results, sixth paragraph. What happens to mutations when the medium is not changed every two weeks? And what does it mean that the medium is changed to "maintain oligo elements" (are these nucleotides?).

When the medium is not changed, the dying cells are releasing nitrogen that can be used to feed the survivor cells. The procedure to change the medium was used by other teams (Sajiki K, Journal of Cell Science 2009). When the medium is not refreshed, especially after two- and four-weeks, the overall viability of the cell population is dropping faster.

The strain used in our study is a prototroph and the medium neither contain nucleotides nor amino-acids, but oligo-elements – trace concentrations of – salts, vitamins and minerals (described in Mitchison, JM Physiological and cytological methods for *S. pombe*. Methods in Cell biology 1970).

4) "Mutations found more than once in clonal population" – do the authors mean "more than once in different clonal populations"?

Thank you for the remark; we corrected the sentence “more than once in a clonal population”. Additionally, several point mutations occurred independently in different clonal populations and were used to estimate the accumulation of mutations at the genome level (Synonymous versus non-synonymous SNV, in Table 4—source data 1).

5) Figure 1. Most mutations are apparently +/-1 indels. Do they arise in homonucleotide runs? I am not sure what "low complexity" sequences means.

Sorry for the misunderstanding. The data of +/-1 indels are shown in Supplementary file 1 (for ura4/5) and Supplementary file 2 (for the genome). For ura4/5 mutants, referring to the reviewer’s comment, we corrected the sentence: “within homonucleotide runs”. The proportion of indels in homonucleotide runs seems more frequent at the genome level (See Supplementary file 2).

6) Results, fourth paragraph. I find statements about evolutionary pressures overstated, based on this small survey of two genes whose nucleotide sequence may – or may not – reflect the entire genome. The small changes in + and – indels are insufficient for such a claim, especially because the 3-month data don't support the claim.In this regard, is this whole genome or exome sequencing? If whole genome, were there no changes in copy number or other changes in the repeated sequences around centromeres, rDNA and elsewhere? These changes would support the idea that nutritional restriction would favor deletion to make a smaller, easier-to-replicate genome, even if under supposedly nonreplicating conditions.

The discussion about human was reduced.

The three-months data have been re-enforced with more genomes and the data analyzed quantitatively enough to conclude that the vegetative and quiescent mutational spectra are different (Essential revisions).

About CNVs, the sequencing was performed on whole genomes (coverage >50x) and repetitive sequences (rDNA, centromeric DNA, mitochondrial DNA) were excluded from the analysis. CNVs into the non-repetitive regions were analyzed using the free algorithm (Boeva et al. Bioinformatics 2011, 27, 268-269)but no CNVs have been detected

7) Figure 2 I am unable to evaluate the corrections to mutation frequencies based on the author's new method of estimating "essential" amino acids. I do not see how they evaluated the better predictive fit of this method relative to Lang and Murray's, which seems to be a straightforward calculation based on the idea that STOP codons become over-represented because many nonsynonymous changes are still functional. Unless the GC content is 50%in coding regions, and given that some SNVs are much more frequent than others, the statement that each mutation has a probability of 1/k may not be correct. In any case please provide the criteria by which the new method outperforms Lang and Murray and better explain the corrected frequencies.

We have corrected and added new information.

8)Results, seventh paragraph. What was the estimated mutation frequency for 3 months?

Sorry, we added into the text:

“Although low, 0.61 mutation per genome after three months, is 3.5-7 times higher than anticipated by our projection (0.084 to 0.17 mutations/genome/three months, depending on the correction method)”

9) Table 3. Is there a way to show which of the 4 or 6 mutations has which of the 8 phenotypes? And, since there was only about 0.7 mutations per genome, if I read correctly, it should be evident – or highly suggestive – what mutation in what gene caused these phenotypes. Do these mutations make sense in terms of their phenotypes?

As explained above the mutations responsible for the strains with phenotype were not included in the 3-month mutational survey. We now have sequenced six of them, they are mapping to *alg5, alg8, aur1, ccc2, gdi1* and *oca8* (dolichyl-phosphate beta-glucosyltransferase, glucosyltransferase -predicted), inositol phosphorylceramide synthase -predicted, copper transporting ATPase -predicted, GDP dissociation inhibitor -predicted and cytochrome b5 -predicted, respectively). Not much is known about these genes (membrane trafficking and cell wall), three are essential. The information was added at the end of the Results section:

“We sequenced the genomes of the six strains isolated after three months and found mutations mapping in *alg5, alg8, aur1, ccc2, gdi1* and *oca8* genes. These genes encode for cell wall and vacuole/membrane trafficking functions that might improve viability during quiescence.”

10) The biggest question surrounding this approach is why these changes in mutation occur. In bacteria, SLAM reflects the increased use of mutagenic DNA polymerase DinB, leading to a change in mutational spectrum. How do bacterial mutation rates change with time? In budding yeast, DNA repair has an 1000-times increase in mutation rate compared to replication, with evidence of the use of a different DNA polymerase in some instances. None of these studies is cited, although Foster's early work on adaptive mutation is. Rosenberg's extensive later work deserves attention. Ditto Strathern's and Haber's yeast work.So, if starving (quiescent) cells are subject to the use of a different polymerase or an altered mechanism of DNA repair, there well could be a change in mutation rate. These possibilities should be explored, perhaps instead of rather unconnected musings on the differences between sperm and ovum.

With respect to the above remarks several changes have been done. (See Essential revisions 1).

We also added in the Introduction the reference of the Rosenberg’s review (Rosenberg, 1997).

Similarly, we added: Holbeck and Strathern, 1997; Hicks, Kim and Haber, 2010;.

Along with gene conversion, nucleotide modification and transposition (Results, second paragraph) we added “meiotic recombination”.

The question on the potential impact of the TLS pathway has been added and discussed (See Essential revisions 2).

The discussion on human along with sperm and ovum are removed. But I’m keeping in mind that “What is true for *E. coli* is true for elephant” (J. Monod).

Reviewer #3:[…] This work addresses a long-recognized but difficult to study, yet important gap of knowledge in mutagenesis, the highly complex question of the stability of genomes in non-replicating cells. The authors approach is based on the well-defined fission yeast model, where nitrogen-starvation can induce a "physiological" quiescent state, as well as of state of the art targeted and genome-wide analyses. The experimental setup has generated highly interesting data that one would like to explore further, both conceptually and mechanistically. While I greatly appreciate the authors efforts to advance this area of research and see the potential importance of the work, my impression is that the state of the current manuscript is somewhat immature; the data is purely descriptive at this point and, rather than using them to develop (and test?) hypotheses regarding the molecular mechanisms promoting mutagenesis in quiescence, the authors focus on genome evolutionary interpretations that seem to stretch the evidence presented considerably. These are definitely interesting considerations but the experimental support is not compelling. In my opinion the manuscript would benefit greatly by the addition of some controls (G0-phase validation), more rigorous analyses and discussion of the actual data, and a more thorough consideration/discussion (perhaps testing) of the potential mechanisms involved in quiescence mutagenesis. Addressing the following questions might help.1) Mutational baseline: I have a number of questions relating to the discrimination of replicating from quiescent cells, i.e. the definition of baseline, which is critical here. (i) It is assumed that a majority of cells enter G0 under nitrogen-starved conditions. While this it probably correct, was the cell cycle status validated in these experiments and where proportions of G0 cells assessed? (i) Vice versa, what would be the proportion of quiescent cells in proliferating populations of S. pombe cells? (iii) How are pre-existing persisters dealt with and excluded? (iv) Why isn't there more fluctuation in mutation frequencies at end of replication, early quiescence? How many cultures were tested?

Thank you for the remarks. We added new data about the quiescence population and fluctuation analysis are discussed. See above the Essential revisions and more details are following.

i) The procedure used in this work is classical and reproducible. Following this procedure, most if not all the cells replicate once and divided twice to enter in G1 and then in quiescence. Each time, the cell number, the cell size and their FACS analysis are carefully determined.

We have introduced microscopic analyses showing the homogeneity of cell size and determined the proportion of cells with septum every day, from 1 to 15 days, during the quiescence time course.

ii, iii) The persisters hypothesis is now discussed.

iv) We have performed fluctuation-like experiments. This is important at the early time point (Day 1) in wild type and even more the pcn1 mutant strain to avoid sequencing the same mutations arising during the pre-cultures. Fluctuation during latter time points in quiescence is less important because we are measuring a mutation accumulation.

Corrections have been added in Results and Discussion.

2) FOA mutation assay and time of occurrence of mutations: The mutation assay described bases on the selection of 5-FOA resistant survivors. FOA itself perturbs the uracil metabolism and, thereby also the dNTP balance. This in itself can generate a shift in mutational spectra. In the experimental setup used, mutations may emerge only after re-plating of cells on FOA media, possibly by repair of replication of DNA damage that was induced during quiescence. Nucleotide imbalance may then affect the type of mutation generated. Was this considered? What is the authors’ take on this? I think, this should be addressed briefly in the Discussion.

Yes, we have been wondering about FOA appearing on the plate. To minimize the problem, we have standardized our FOA plates and only incubate the cells for three days. After five days, small secondary FOA^R^ colonies appear. These mutants are more likely arising post plating.

To evaluate the potential bias imposed by the FOA we compared the FOA^R^ mutations obtained after directly plating cells from quiescence or after one cell division prior plating cells onto FOA containing plates. The mutations appearing in pairs exhibits the quiescence spectrum that is supporting that FOA is equally affecting the growing and quiescence mutational spectrum (See Essential revisions). Thank you for motivating us to compare the FOA^R^ mutations arising prior and after DNA replication.

3) The FOA-based mutation assay is not fully unbiased as it scores functional changes related to uracil mechanism only. This limitation should be phrased in the description of the assay. To what extent could the spectra shift be explained by the FOA effect on nucleotide metabolisms (changes involving T?).

Yes, we agree. We also added the limitation of the FOA-based assay (Essential revisions 3).

4) Mutation spectra: I suggest displaying the frequencies of individual single-nucleotide changes (e.g. Figure 1, respective data in Table 1 and Table 2) as normalized to the total number of relevant scorable base pairs. This would provide mora accurate information about true mutation preferences.

We have represented the frequencies of individual single-nucleotide changes as normalized to the total number of relevant scorable base pairs and did not see any significant difference with that of the observed data in Figure 2. Therefore, we have left the Figure 2 untouched and added a reference to Figure 2—figure supplement 1 that shows both graphs together with a table of the estimated values of the relevant scorable base pairs in *ura4* and *ura5*.

Figure 2—figure supplement 1: (A) Mutation spectrum established on 105 SNVs found in the *ura4*^+^ and *ura5*^+^ genes at day 1 and 146 at days 8 + 15 (from Figure 1). (B) Same as in A, but normalized to the total number of relevant scorable base pairs. (C) Counts of all possible STOP and nonsense mutations for each substitution in *ura4* and *ura5.*

5) CpG mutagenesis: It is interesting, very interesting indeed, that CpG dinucleotides are preferential sites of mutation. Again, it would be interesting to see if and to what extent the normalized frequency (normalized to total number of such sites in the scorable sequences) is higher than predictable (statistics against all dinucleotides). This is likely, given the coding importance of these nucleotides and the fact that they not only preferentially mutate by C (or 5mC) deamination but also by alkylation of G (6-methylguanine), for instance.

The CpG bias cited a day 1, although interesting, is a comparison with the finding of the Lynch and Hall laboratories and not the purpose of this manuscript. We have updated the figure to show the normalized frequency of mutations in each dinucleotide for each gene compared to the frequency of occurrence of every dinucleotide in the respective open reading frame. The differences between the two reporter genes make it difficult to draw a strong conclusion.

We have modified the sentence in the text and modified the figure legend.

“Interestingly, as previously observed {Behringer and Hall, 2015; Farlow et al., 2015; Zhu et al., 2014} the CpG dinucleotides are found among the top mutated dinucleotides in *ura4* and *ura5* (Figure 2—figure supplement 2)”.

Figure 2—figure supplement 2:Normalized percentage of each dinucleotide in the open reading frames of *ura4*^+^ and *ura5*^+^ after one day in quiescence. The percentage of every dinucleotide mutated at day one is compared to the frequency of occurrence of the same dinucleotide in the respective open reading frame.

6) Statistics in general: Are the differences in spectra between day 1 and day 8+15 statistically significant (Figure 1 and Figure 2)?

We added statistic values. In particular, for days 8+15:

“Days 8 and 15 are not statistically different for the Ts:Tv and AT bias (chi2 test) and were therefore combined. Day 1 is statistically different from days 8+15 (Ts/Tv: P<0.05; AT bias: p<0.05 by chi2 tests).”

7) Normalizations: I would be interested to have the analysis performed in Figure 2 extended for individual SNVs (again normalized to the total number of scorable bases in each category).

See point 4 about the differences between *ura4* and *5*.

8) Genome-scale mutation assessment: This precious dataset should, in my opinion, be explored further, for instance to validate the CpG bias, or sequence context biases, or the FOA bias. I could imagine there is quite some potential there. Also, it seem important to me to compare this dataset to one of exponentially growing cells (same number of cell after similar time of continuous cultivation). In my opinion, this experiment is a real strength of the paper.

We extended the genome wide sequencing but the number of SNVs (and CpG) is not high enough.

9) Discussion: The Discussion is interesting and stimulating but rather far reaching. Given that some caveats exist (FOA, persisters…) and the datasets are not exhaustively explored and also entirely descriptive, I would prefer to have some discussion of the actual data, their strengths and weaknesses, and the mechanisms of mutagenesis implicated. What could be the molecular reason for the spectrum shift?

We introduced new data in the Results section and discussed further the notion of persisters and the Chronos hypothesis and removed some of the human part.

[Editors' note: further revisions were requested prior to acceptance, as described below.]

The manuscript has been improved but there are some remaining issues that need to be addressed before acceptance, as outlined below:Please acknowledge that repair processes could indeed give 1000x increases that you see.

Initial version:

“If these replicating cells were generating the FOA^R^ mutants observed during quiescence, we will have to conclude that they generate 1,000 times more mutations per replication. Furthermore, these rare replicating cells will have to divide at a constant rate during time in quiescence. However, the number and the distribution of mutations per genome observed in the three months aged cells is hardly compatible with this hypothesis. Therefore, we favor the Chronos model where DNA lesions that are arising during quiescence [Mochida and Yanagida, 2006; Ben Hassine and Arcangioli, 2009] are repaired with errors that generate mutations with time.”

New version:

“If these replicating cells were generating the FOA^R^ mutants observed during quiescence, we will have to conclude that they generate 1,000 times more mutations per replication. Double-strand break repair is a process that could generate this level of mutations [Hicks, Kim and Haber, 2010; Holbeck and Strathern, 1997]. Nevertheless, these rare replicating cells will have to divide at a constant rate during time in quiescence. Additionally, the number and the distribution of mutations per genome observed in the three months aged cells is hardly compatible with this hypothesis. Therefore, we favor the Chronos model where DNA lesions (not excluding DSBs) that are arising during quiescence [Mochida and Yanagida, 2006; Ben Hassine and Arcangioli, 2009] are repaired with errors that generate mutations with time.”

We also added two recent references at the beginning of the Discussion, on RNAi and telomere, which explore the genetic of quiescence.

“Two recent studies on RNA interference and telomere stability highlighted the importance of quiescence for chromosome biology (Maestroni et al., 2017; Roche, Arcangioli and Martienssen, 2016).”